# Estimating the cost of illness of acute Japanese encephalitis and sequelae care in Vietnam and Laos: A cross-sectional study

An Le Thanh Nguyen[1]*, Rose Slavkovsky[2], Hai Thanh Phan[3], Huong Thi Thu Nguyen[4], Souphaphone Vannachone[5], Dang Hai Le[4], Audrey Dubot-Pérès[5,6,7], Manivanh Vongsouvath[5], Son Thai Dinh[3], Anthony A. Marfin[2], G. William Letson[2], Huong Minh Vu[8], Dung Chi Tham[9], Mayfong Mayxay[5,7,10], Elizabeth A. Ashley[5,7], Thai Quang Pham[4], Clint Pecenka[2]

1 PATH, Ho Chi Minh City, Vietnam, 2 PATH, Seattle, Washington, United States of America, 3 School of Preventive Medicine and Public Health, Hanoi Medical University, Hanoi, Vietnam, 4 National Institute of Hygiene and Epidemiology, Hanoi, Vietnam, 5 Lao-Oxford University-Mahosot Hospital-Wellcome Trust Research Unit (LOMWRU), Microbiology Laboratory, Mahosot Hospital, Vientiane, Lao People's Democratic Republic, 6 Unité des Virus Émergents (UVE: Aix-Marseille Univ-IRD 190-Inserm 1207), Marseille, France, 7 Centre for Tropical Medicine and Global Health, Nuffield Department of Medicine, University of Oxford, Oxford, United Kingdom, 8 World Health Organization Viet Nam, Hanoi, Vietnam, 9 PATH, Hanoi, Vietnam, 10 Institute of Research and Education Development, University of Health Sciences, Vientiane, Lao People's Democratic Republic

* altnguyen@path.org

## Abstract

### Background

Japanese encephalitis (JE) is a leading cause of acute encephalitis syndrome and resulting neurological disability in Asia and the Western Pacific. This study aims to estimate the cost of acute care, initial rehabilitation and sequelae care, in Vietnam and Laos.

### Methodology

We conducted a cross-sectional retrospective study using a micro-costing approach from the health system and household perspectives. Out-of-pocket direct medical and non-medical costs, indirect costs, and family impact were reported by patients and/or caregivers. Hospitalization costs were extracted from hospital charts. Acute costs covered expenditures from pre-hospital to follow-up visits while sequelae care costs were estimated from expenditures in the last 90 days. All costs are in 2021 US dollars.

### Principal findings

242 patients in two major sentinel sites in the North and South of Vietnam and 65 patients in a central hospital in Vientiane, Laos, with laboratory-confirmed JE were recruited regardless of age, sex, and ethnicity. In Vietnam, the mean total cost was $3,371 per acute JE episode (median $2,071, standard error [SE] $464) while annual costs were $404 for initial sequelae care (median $0, SE $220) and $320 for long-term sequelae care (median $0, SE $108). In Laos, the mean hospitalization costs in acute stage were $2,005 (median $1,698, SE $279)

**Data Availability Statement:** All relevant data are within the paper and its Supporting Information files.

**Funding:** The study is funded by the Bill & Melinda Gates Foundation (ID#: AWD-252768/OPP1115522/INV-008523; Recipient: PATH). The funders had no role in study design, data collection and analysis, decision to publish, or preparation of the manuscript.

**Competing interests:** The authors have declared that no competing interests exist.

and the mean annual costs were $2,317 (median $0, SE $2,233) for initial sequelae care and $89 (median $0, SE $57) for long-term sequelae care. In both countries, most patients did not seek care for their sequelae. Families perceived extreme impact from JE and 20% to 30% of households still had sustained debts years after acute JE.

## Conclusions

JE patients and families in Vietnam and Laos suffer extreme medical, economic, and social hardship. This has policy implications for improving JE prevention in these two JE-endemic countries.

## Introduction

Japanese encephalitis (JE) is a leading cause of acute encephalitis syndrome (AES) and resulting neurological disability in Asia and the Western Pacific, with an estimated 67,900 new cases of JE occurring annually [1, 2]. The disease is caused by JE virus—a mosquito-borne flavivirus. Among cases of JE-attributable AES, as many as 30% are fatal, and long-term neurological or physical sequelae occur in 30% to 50% of survivors [1, 3, 4]. Approximately 75% of JE cases occur in children younger than 15 years [2]. There is no cure for JE, and treatment is largely supportive to relieve symptoms [1]. JE vaccines have been used for military since the Second World War, and the World Health Organization (WHO) prequalified the first JE vaccine in 2013. Four JE vaccines are now WHO prequalified [1, 5]. Data from JE vaccination and surveillance programs demonstrate significant reductions in JE incidence following the implementation of JE vaccine, particularly when vaccine coverage is high [6].

In Vietnam, thousands of viral encephalitis cases are recorded annually in the Infectious Diseases Surveillance System; JE is a main cause of acute central neural system infection [7]. Annual JE incidence among children aged 0 to 14 years is estimated to be 4.7 per 100,000 [2]. Between 2010 and 2019, an average of 252 new JE cases were reported annually among all age groups [8]. JE transmission is most prevalent in June and July, with the highest JE incidence in children aged 5 to 14 years [9]. The inactivated mouse brain–derived JE vaccine (JEVAX) manufactured in Vietnam has been used in the Vietnam Expanded Program on Immunization (EPI) since 1997. By 2015, 100% of communes of Vietnam had deployed JEVAX with a three-dose regimen for children aged 1 to 5 years [10]. Since vaccine introduction, the proportion of JE-positive cases among viral encephalitis cases has fallen from nearly 50% and now hovers near 10% [10].

In Laos, the annual incidence of JE among children younger than 15 years is estimated to be 10.6 per 100,000 [2]. Only one central hospital (Mahosot Hospital in the capital, Vientiane) offers laboratory diagnosis for JE virus infection by polymerase chain reaction (PCR) and enzyme-linked immunosorbent assay (ELISA) on cerebrospinal fluid (CSF) and serum specimens. Laos conducted its first JE vaccination campaign in 2013 using a vaccine called CD-JEV and, in 2015, introduced JE into the nationwide routine immunization program. However, vaccine uptake declined rapidly after initial introduction [11–13].

Despite a WHO recommendation, multiple prequalified products, and proven positive impact of JE vaccine on disease incidence, as of 2020, only 15 of the 25 countries with JE transmission risk (Australia (Outer Torres Strait Islands), Malaysia (Sarawak), Japan, Republic of Korea, Thailand, Cambodia, Laos, Myanmar, Indonesia (Bali), Philippines (three high-incidence regions), China including Taiwan, India (about 40–50% of districts), Nepal, Sri Lanka, and Vietnam) have a national or subnational JE immunization program. Some countries,

including Democratic People's Republic of Korea and Brunei, have introduced JE vaccine through campaigns without establishing it in routine immunization [13]. Additionally, as many immunization programs have been disrupted by COVID-19, and some are facing financing pressures, [14, 15] it is useful to consider the health and economic consequences of JE vaccination.

A previous study in Vietnam assessed the physical and neurological disability among JE survivors at two provincial hospital surveillance sites, using the Liverpool Outcome Score at least four months following hospital discharge. Among the 26 patients assessed, 8% had severe sequelae, 19% had moderate sequelae, and 31% had mild sequelae. The most frequent sequelae among all survivors were behavioral problems [16]. On the economic side, a 2017–2018 study on direct medical expenditures of acute viral encephalitis patients at health facilities in three North-West provinces of Vietnam showed the average direct medical cost was approximately 8 million Vietnamese dong (~US$350). Patients paid 5% of total expenditures with the remainder paid by the government via social health insurance [17]. Although this study identified the extreme financial cost of an episode of JE, it does not provide insight into longer term and potentially more economically devastating costs associated with JE sequalae.

In Laos, a hospital-based study prospectively followed JE virus–infected patients using the Liverpool Outcome Score to assess sequelae at follow-up. Among survivors, 61.2% had neurological sequelae. Moderate and severe sequelae were significantly higher in children [11]. Similarly, although this study illuminates the long-term physical impact of JE, especially on children, there remains a lack of evidence on the economic burden of JE and resulting sequelae in Laos. A major source of healthcare financing in Laos is out-of-pocket expenditure. Even with national health insurance, patients still need a co-pay of 25% if the treatment has high cost (exceeding $600), understanding the cost of the illness emphasizes the need to prevent JE infection and avoid potential catastrophic expenditure [18].

As such, this study seeks to complement prior research by estimating the cost of acute care for JE, as well as the cost of initial rehabilitation and sequelae care, in Vietnam and Laos. The results fill a research gap and identify costs of JE illness and the initial and long-term rehabilitation and sequelae care. This information is intended to underscore the value of JE vaccination to inform country-level policy and decision-making about JE vaccine introduction, renewed commitment, and sustainability.

## Methods

This retrospective study estimates the cost of acute JE illness and its sequelae using a micro-costing approach. Data were obtained through review of hospital administrative data and interviews with families. This study evaluates costs from health system and household perspectives, together representing a modified societal perspective. Our goal was to use similar methods in the two study sites, but it became clear that the study team would have to adapt the Vietnam study methods to the Laos settings in the ways described below.

### Study area and population

Vietnam has eight national sentinel sites for JE surveillance, two of which were selected as centers for recruitment: Vietnam National Children's Hospital (VNCH) in Hanoi and Pasteur Institute in Ho Chi Minh City. These sites represent different geographical areas and approximately 52% of JE cases from around the country [10]. Researchers examined databases at the two sites to identify eligible participants. Prospective participants were grouped by province and provincial subregions. The study team began contacting potential participants in the province with the largest case numbers and continued to reach the target sample size.

The study sought to understand the costs of JE treatment during three phases: acute JE, initial rehabilitation and sequelae care, and long-term rehabilitation and sequelae care. Eligibility for the acute care group included AES patients with JE virus infection confirmed by ELISA or PCR diagnostic test who were discharged within 12 months prior to enrollment. The initial and long-term sequelae groups included those with laboratory confirmed JE-attributable AES within 3 to 24 months and 25 to 120 months from discharge to enrollment, respectively. Individuals included in the initial rehabilitation and sequelae group could also be included in the acute care group. Inclusion criteria consisted of willing patients with laboratory confirmed JE-attributable AES within the specified period and/or their parents/caregivers. Exclusion criteria included those who did not consent, those that relocated during the study, patients younger than 18 years without a consenting caregiver, and those with known previously identified health challenges (e.g., cancer, HIV) that could inhibit ability to differentiate cost of care.

In Laos, the study team examined the Mahosot Hospital database. We acknowledged the limitation of sourcing the cases from the tertiary hospital database, which may mean that the severity level may not be reflective of JE severity in the community; however, it is the only facility to provide laboratory-confirmed JE cases. The study team identified and contacted potential participants for recruitment until reaching the target sample size or running out of cases. There was not an adequate number of eligible cases in the database for the acute and initial sequalae groups, so the study team dropped the acute group to focus on the cost of initial and long-term sequelae care in Laos. The team did attempt to obtain hospitalization costs in the acute stage whenever available.

## Sample sizes

The study team used the following formula to estimate sample sizes for the acute and two sequelae groups:[19]

$$n = ceiling \left[ \left( \frac{precision^2}{CV^2 \times Z_{1-\alpha/2}^2} + \frac{1}{N_0} \right)^{-1} \right]$$

Precision is set at +/- 10% of the mean cost, with coefficient of variation assumed to be 0.5. Considering the estimated 252 annual JE cases per year since 2011 and Vietnam case fatality rate of 12%, the estimated total sample size is 240 in Vietnam [7, 10]. According to the Mahosot Hospital database, there are 20 new laboratory-confirmed JE attributable AES cases per year in Laos. With the estimated mortality rate of 18%, [11] the total estimated sample size for the initial and long-term sequelae group is 79 in Laos. The number of cases in each study group are presented in S1 Table.

## Data collection

Data collection occurred between November 2021 and April 2022. Data collection procedures varied slightly between the acute and sequelae groups, although patient/caregiver consent was the initial step in all cases.

In the acute costs group in Vietnam, study participants were interviewed using a structured questionnaire covering basic demographic information, socioeconomic status, direct medical and non-medical costs, and indirect non-medical costs incurred before, during, and up to 90 days after hospitalization. Study staff also undertook a review of hospital records to extract the direct medical costs associated with each case. The severity of each JE illness was captured by the Glasgow Coma Score (GCS) or a doctor's note on severity if GCS was not available in the patient's hospital records.

**Table 1. Cost categories and example costs included in the Japanese encephalitis (JE) cost of illness study.**

| Direct medical costs | | Unit of measurement |
|---|---|---|
| Medicines | Prescription and over-the-counter drugs, homeopathic medicines, and traditional medicines such as herbs | Vials, tabs, packets, bottles |
| Hospital and rehabilitation facility-based fees | Registration, administration, consultation, and hospital bed | Counts, days |
| Provider-based costs | Consultation by doctors, nurses, physical therapists, community health workers, and other traditional health care providers (acupuncturists) | Service counts |
| Diagnostics/lab fees | JE diagnostic tests (ELISA, PCR), imaging (CT, MRI, ultrasound), monitoring tests (CSF analysis, metabolic panel, etc.) | Test counts |
| Procedures/interventions | Oxygen, lumbar puncture, humidifier, physiotherapy, etc. | Procedure counts |
| **Direct non-medical costs** | | |
| Travel to and from heath care facility | Bus or taxi fare, fuel for personal vehicle, etc. | Number of trips |
| Lodging and meals | Lodging and meals for patient and caregiver(s) while seeking care outside home | Nights, number of meals |
| Miscellaneous costs | Telephone calls, gifts from visitors | Counts |
| **Indirect non-medical costs** | | |
| Opportunity cost of time spent caring for illness | Lost wages for time spent by families accompanying the patient to health care facilities and providing direct care to patients | Days |
| Productivity loss of adults with illness and disability | Lost wages due to illness and impairment | Days |

Participants in the initial and long-term sequelae groups in Laos and Vietnam started with a phone screen by study staff to assess the severity of their sequelae using the Liverpool Outcome Score (LOS), a validated tool to assess outcome of patients following encephalitis [20]. Individuals were recruited to participate in the study if they had some level of sequelae, indicated by a score of less than 5 in the first ten questions of the LOS. The presence of sequelae then triggered a home visit in which trained study staff observed the patient performing simple motor tasks during the field interview. The full assessment took between 10 and 40 minutes per patient [16, 20, 21]. In addition, participants were interviewed using a structured questionnaire as described above. In cases where the patient was receiving ongoing care from a health facility, such as a rehabilitation center or physical therapy clinic, in the 3 months prior to the interview, study staff tried to retrieve receipts, where feasible, to extract direct medical costs. Table 1 illustrates the cost categories examined in the study. For each cost item, we collected the unit, quantity, and charge per unit to calculate the cost of each item for each patient.

The questionnaires also sought information to assess the impact of JE-related expenses on household financial wellbeing and to explore health care financing strategies of coping with the illness. Questions examined the decisions families made to care for the patient and to cover health care costs, and how the illness generally impacted the family.

All costs were collected in 2021–2022 Laotian kip (LAK) or Vietnam dong (VND) as costs for the acute group occurred in 2021 and initial and long-term sequelae groups were asked for costs occurring in the last 3 months. Hospitalization costs in Laos were converted to 2021 Laotian kip using the consumer prices index from the World Bank 2010–2021 data [22]. Costs were converted to 2021 US dollar (USD) using the annual average official exchange rate (1 USD = 9,697.92 LAK = 23,159.78 VND) [23].

## Analysis

Costs were calculated using quantities multiplied by per unit costs (e.g., number of visits multiplied by cost per visit). For direct medical costs, unit costs were charges from providers, which

we assumed to cover overhead costs. For direct non-medical costs, unit costs were the amount paid by the families. For indirect costs, unit costs were the daily totals of official and unofficial income lost from patients and household members who missed earning each day.

We calculated the total cost per patient for acute JE by adding up all costs from pre-hospital to in-hospital, discharge, and follow-up visits up to 90 days after discharge. For initial and long-term sequelae and rehabilitation care, we estimated the annual cost per patient using all costs that occurred in the last 90 days.

### Ethics

Ethical approval was provided by the Western Institutional Review Board and Copernicus Group (WCG IRB—IRB Pr. No. 20213999) and the National Institute of Hygiene and Epidemiology Institutional Review Board in Bio-medical Research for the study in Vietnam (IRB-VN01057/IORG 0008555), and by the WCG IRB (IRB Pr. No. 20215005), Oxford Tropical Research Ethics Committee (OxTREC Ref. 559–21), and University of Health Sciences Research Ethics Committee (No. 280/REC) for the study in Laos. Written consent was obtained from patients and/or parents, caregivers, or guardians of the minors in the initial step of data collection.

## Results

### Participant characteristics

In Vietnam, the study included 242 participants (55 acute, 88 initial rehabilitation sequelae, and 99 long-term sequelae). Male patients were the majority at 67%. The age of patients when they fell ill from JE varied from 0 to 78 years with the mean of 8.6 years old (median 7.0, standard error [SE] 0.6). Most patients lived in rural areas (89.7%), with a family size of 4.8 people (SE 0.1) with 1.9 income earners on average (SE 0.1). The average household monthly income was about $600 (median $388, SE $59) and monthly expenses were $423 (median $338, SE $25). On average, these households spent about $31 per month for health care (median $2, SE $6). Nearly half of patients (49.2%) were not vaccinated or had not completed the full three-dose regime of JE vaccination, and 21.9% did not know their or their child's vaccination status. The mean age of those fully vaccinated (28.9% of all cases) was 6.9 years old. According to the EPI vaccination schedule, the full JE vaccine regimen is complete by age 2, which means the patients included in this study acquired JE a mean of 4.9 years after finishing their last dose. For the acute group, 41.8% have moderate to severe JE according to reported Glasgow Coma Score or doctor's assessment in medical records. For sequelae groups, 42.8% were assessed to have moderate sequelae and 17.7% had severe sequelae using LOS. The average LOS outcome score of patients in both initial and long-term sequelae groups was 3.26 (SE 0.06), which means there were effects on physical function, personality, or need for medication, but on average, patients could still have independent living.

In Laos, the study included 65 participants (9 initial rehabilitation sequelae and 56 long-term sequelae) with male patients comprising 74%. The age of patients when they acquired JE varied from 1 to 55 years old with the mean of 13.7 years old (median 11.0, SE 1.3). Most patients lived in rural areas (44.6%) and peri-urban areas (41.5%). The average household monthly income was $525 (median $361, SE $80) and expenses were $450 (median $300, SE $50), of which about $38 per month were for health care spending (median $8, SE $19). Most patients in the study did not get JE vaccine (78.5%) and 21.5% did not know their vaccination status. Among those assessed to have sequelae at the site visit (57/65), 49% had moderate and 10.8% had severe sequelae. The average LOS outcome score was 3.42 (SE 0.10), which has similar interpretation to patients in Vietnam (the score of 5 means full recovery).

**Table 2. Patient and household characteristics.**

| | VIETNAM | | | LAOS | |
|---|---|---|---|---|---|
| | Acute (n = 55) | Initial sequelae (n = 88) | Long-term sequelae (n = 99) | Initial sequelae (n = 9) | Long-term sequelae (n = 56) |
| Female, n (%) | 16 (31.3%) | 29 (32.9%) | 34 (34.3%) | 2 (22.2%) | 15 (26.8%) |
| Age in years at JE onset, mean (median, SE) | 8.9 (8.0, 1.0) | 10.0 (8.0, 1.3) | 7.2 (6.0, 0.7) | 14.1 (7.0, 5.6) | 13.6 (12.0, 1.3) |
| Rural area, n (%) | 50 (90.9%) | 82 (93.2%) | 85 (85.9%) | 5 (55.6%) | 24 (42.9%) |
| Household size, mean (SE) | 4.9 (0.2) | 4.9 (0.2) | 4.7 (0.1) | 6.7 (0.9) | 6.2 (0.3) |
| Average household income earners (SE) | 1.9 (0.1) | 1.9 (0.1) | 1.9 (0.1) | 2.3 (0.2) | 2.2 (0.1) |
| Average monthly income in USD (median, SE) | $762 ($443, $137) | $470 ($345, $52) | $631 ($389, $114) | $425 ($309, $146) | $541 ($361, $90) |
| Average monthly expenses in USD (median, SE) | $486 ($366, $60) | $330 ($314, $24) | $471 ($366, $45) | $603 ($305, $226) | $426 ($293, $45) |
| Average monthly expenses in USD for health care (median, SE) | $56 ($0, $25) | $25 ($1, $8) | $23 ($3, $6) | $185 ($9, $136) | $15 ($5, $4) |
| JE vaccine status (%) | | | | | |
| Not at the age of vaccination | 4 (7.3%) | 0 | 7 (7.1%) | - | - |
| 0 dose | 16 (29.1%) | 25 (28.4%) | 22 (22.2%) | 7 (77.8%) | 44 (78.6%) |
| 1 dose | 4 (7.3%) | 8 (9.1%) | 6 (6.1%) | - | - |
| 2 doses | 4 (7.3%) | 13 (14.8%) | 10 (10.1%) | - | - |
| 3 doses | 16 (29.1%) | 26 (29.5%) | 28 (28.3%) | - | - |
| Unknown | 11 (20.0%) | 16 (18.2%) | 26 (26.3%) | 2 (22.2%) | 12 (21.4%) |
| Severity of JE illness (acute group) and sequelae (initial and long-term groups) | | | | | |
| Mild | 32 (58.2%) | 30 (34.9%) | 37 (39.4%) | 3 (37.5%) | 15 (30.6%) |
| Moderate | 19 (34.5%) | 38 (44.2%) | 42 (44.6%) | 5 (62.5%) | 27 (55.1%) |
| Severe | 4 (7.3%) | 18 (20.9%) | 15 (16.0%) | 0 | 7 (14.3%) |
| Mean LOS outcome (SE) | - | 3.18 (0.08) | 3.32 (0.08) | 3.56 (0.24) | 3.39 (0.12) |

*Note: The frequency of sequelae severity of initial and long-term groups is based on cases with sequelae. A few cases in these two groups had no sequelae (7 cases in Vietnam and 8 cases in Laos), but they didn't reflect non-sequelae frequency as we purposely selected sequelae cases in this study to explore the costs for sequelae care.

Abbreviations: JE, Japanese encephalitis; LOS, Liverpool Outcome Score; SE, standard error; USD, United States dollar.

More details on the participants' characteristics by different study groups in two countries are included in Table 2.

## Cost of JE illness and sequelae

**Vietnam.** The total cost of the acute stage, from pre-hospitalization to 90 days after discharge, was an average of $3,371 (median $2,071, SE $464) in 2021 US dollars. For JE sequelae care, the average annual total costs were $404 (median $0, SE $220) for patients at initial sequelae stage (first two years) and $329 (median $0, SE $109) for long-term sequelae (from year 3 to 10). As the majority of patients (approximately 75%) did not seek care for their sequelae status, the annual total costs were higher among those accessing care as shown in Tables 3 and 4 below. In acute stage, about 72% of the total costs were paid out-of-pocket, while in sequelae stage, the out-of-pocket portion went up to 91% to 93% of the total.

Fig 1 depicts the total and out-of-pocket costs in acute stage and those costs per year in sequelae stage.

Direct medical costs constitute the largest expense at $1,863 for acute stage of all patients, $789/year for initial rehabilitation sequelae, and $660/year for long-term sequelae for patients who accessed care. Table 3 gives details of medical costs, showing drug cost to be the most

**Table 3. Direct medical costs for patients who accessed care in Vietnam.**

| Mean (n, median, SE)* | Acute (n = 55) | Initial sequelae (n = 88) | Long-term sequelae (n = 99) |
|---|---|---|---|
| Total direct medical costs | $1,863 (55, $1,091, $331) | $789/year (23, $86, $472) | $660/year (23, $242, $249) |
| Drug cost | $623 (55, $315, $113) | $175/year (13, $86, $77) | $701/year (11, $104, $439) |
| Diagnostic test cost | $303 (55, $208, $37) | - | - |
| Procedure/intervention/therapy cost | $240 (55, $39, $56) | - | $1,684/year (2, $1,684, $1,425) |
| Facility/provider fees | $465 (55, $256, $71) | $1,696/year (8, $73, $1,287) | $266/year (12, $225, $65) |
| Medical device or equipment | $1,943 (6, $184, $1,788) | $207/year (3, $276, $106) | $155/year (3, $173, $60) |
| Acupuncture/traditional medicine/homeopathy | $133 (4, $108, $67) | $564/year (3, $397, $239) | $216/year (2, $216, $216) |
| Other costs | $11 (53, $0, $6) | - | - |

*All mean costs in the table are based on number of cases indicated first in the parentheses. Abbreviation: SE, standard error.

Non-medical costs such as transportation, lodging, and meals account for the smallest portion of total JE care costs, varying from 17% in initial sequelae group to 19% in acute group and 22% in long-term sequelae group.

common cost for patients. All patients with drug costs of more than $104/year in sequelae groups have severe sequelae conditions.

Households lost an average of 64.2 working days to seek care or take care of patients during the acute JE stage (median 30, SE 12.6 days), with an average income loss of $1,014 (median $389, SE $228) in addition to all other expenses. Patients missed 61.6 days of school (median 18, SE 15.8 days), while all caregivers (siblings, parents, etc.) missed approximately 73.2 days of school in total (median 30, SE 21.2 days). Patients and caregivers of initial and long-term sequelae groups did not miss school days but lost income at an average of $166/year (median 0, SE $95) and $151 (median 0, SE $80) for initial and long-term sequelae care.

Families usually borrowed money, with corresponding interest rates, to cover JE-related expenses. In the acute group, 60% of families borrowed an average of $4,029 including interest. Thirty percent of families in the initial sequelae group had a loan, with interest, of $3,842, and 18% of families in long-term sequelae group had a larger loan of $8,720 due to longer time paying down interest. JE-related expenses from the acute stage caused a prolonged debt to 44.3% of families in the first two years after discharge with repayment of $858/year and in 18.2% of families from 3 to 10 years after discharge with repayment of $597/year. Other responses to cover the expenses include using savings, cutting down other expenses, and selling assets (from jewelry to real estate). Most respondents (73%) perceived that JE-related expenses had a large to extreme impact on the family; 69% said income of members in the household was affected in a large to extreme way; and 81% admitted large to extreme familial stress and fear due to the disease.

**Table 4. Direct non-medical costs for patients who accessed care in Vietnam.**

| Mean (n, median, SE)* | Acute (n = 55) | Initial sequelae (n = 88) | Long-term sequelae (n = 99) |
|---|---|---|---|
| Total direct non-medical costs | $494 (55, $333, $96) | $198/year (14, $65, $104) | $142/year (11, $35, $75) |
| Transportation | $265 (55, $138, $89) | $140/year (13, $43, $71) | $64/year (10, $30, $19) |
| Telephone | $16 (43, $9, $4) | $24/year (4, $13, $15) | $11/year (3, $4, $8) |
| Accommodation | $111 (32, $76, $19) | $138/year (1, $138, -) | - |
| Meals | $149 (52, $122, $26) | $242/year (3, $311, $69) | $17/year (1, $17, -) |
| People support providing other care/tasks | $108 (3, $65, $78) | - | - |
| Other costs | $19 (2, $19, $11) | - | - |

*All mean costs in the table are based on number of cases indicated first in the parentheses. Abbreviation: SE, standard error.

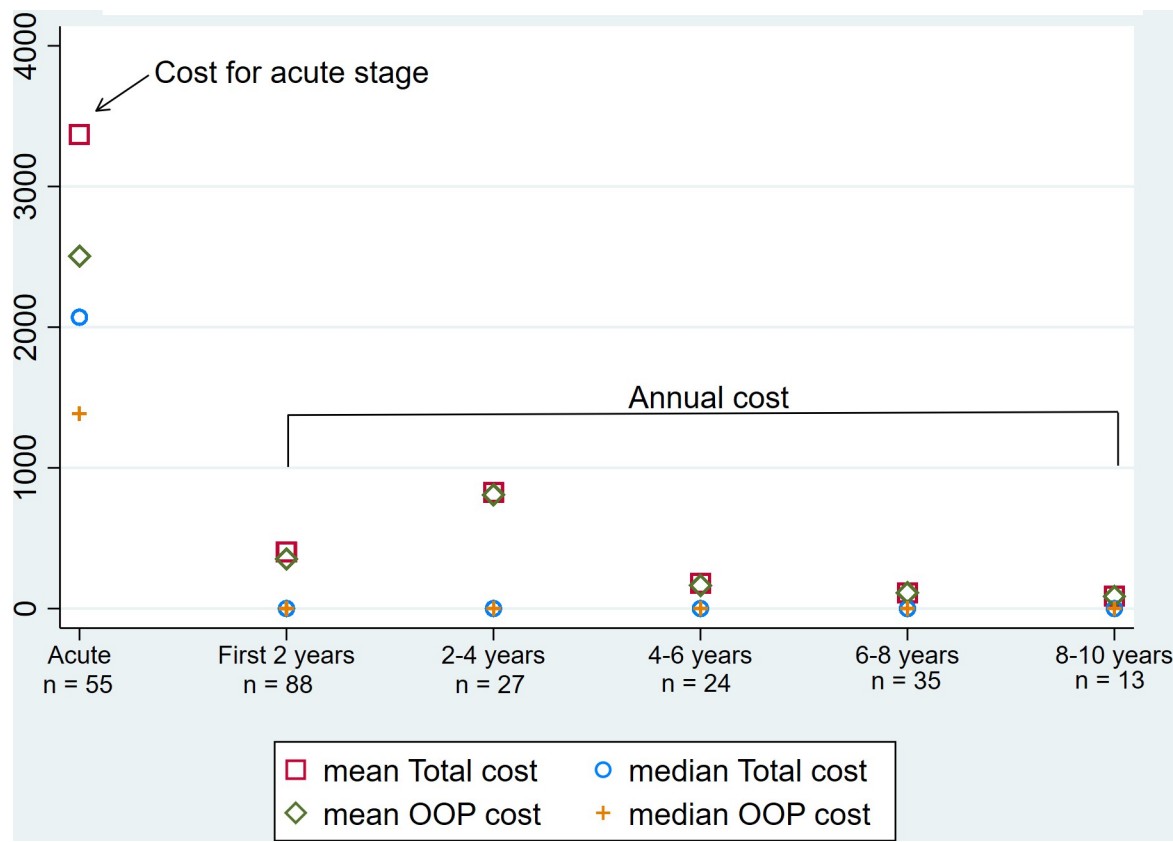

**Fig 1. Total and out-of-pocket costs for Japanese encephalitis (JE) in acute stage and up to 10 years after discharge in Vietnam.**

**Laos.** The total hospitalization cost for acute JE in 24 patients with available data (out of 65 patients included in the study) was $2,005 (median $1,698, SE $279). Almost all patients (23/24) paid an average of 86.6% of the total amount out-of-pocket. Less than one-third of patients (29%) had public insurance, and on average, insurance covered 47.5% of the total amount.

For JE sequelae care, the average annual total cost was $398 (median $0, SE $313), but it is noticeable that only 9.2% of patients accessed health care services even though 87.7% had some level of sequelae. This low rate of accessing sequelae care does not seem to be related to the COVID-19 pandemic, as only 12.3% of patients sought care in the pre-pandemic period. For patients seeking sequelae care (1 mild, 5 moderate, 1 severe sequelae), the annual direct medical cost was $3,586 (median $557, SE $2,832) for outpatient visits, medicines, physical therapy, acupuncture, or crutches. Direct non-medical costs for patients and caregivers were $1,086 per year (median $711, SE $490) for transportation, meals, and phone costs. Breakdown of costs for patients who accessed care in Laos is presented in Table 5.

Although the participants did not report loss of income due to JE-related issues, several intangible costs could be observed. Each of the seven patients with severe sequelae stayed at home and five needed close monitoring by a caregiver, which caused productivity loss of two members in each household. Among 32 patients with moderate sequelae, 27% of patients younger than the age of 18 and 35% older than the age of 18 could not go to school or work. In 12.3% of cases, a caregiver left their paying job to stay at home and provide care to the patient. Even when some families chose not to seek care for sequelae conditions, 20% of them still needed to pay for debt sustained from acute JE, with an average annual payment of $996 (median $444, SE $357). Most respondents assessed that the level of JE impact on their families

**Table 5. Direct medical and non-medical costs for patients who accessed care in Laos.**

| Mean (n, median, SE)* | Initial sequelae | Long-term sequelae | All |
|---|---|---|---|
| **TOTAL DIRECT MEDICAL COSTS** | $17,633/year (1, $17,633, -) | $777/year (5, $242, $249) | $3,586/year (6, $557, $2,832) |
| Drug cost | - | $153/year (3, $87, $79) | $153/year (3, $87, $79) |
| Diagnostic test cost | - | - | |
| Procedure/intervention/therapy cost | $7,424/year (1, $7,424, -) | | $7,424/year (1, $7,424, -) |
| Facility/provider fees | - | $148/year (1, $148, -) | $148/year (1, $148, -) |
| Medical device or equipment | $309/year (1, $309, -) | - | $309/year (1, $309, -) |
| Acupuncture/traditional medicine/homeopathy | $9,280/year (1, $9,280, -) | - | $9,280/year (1, $9,280, -) |
| **TOTAL DIRECT NON-MEDICAL COSTS** | $1,609/year (2, $1,609, $928) | $563/year (2, $563, $179) | $1,086/year (4, $711, $490) |
| Transportation | $1,547/year (2, $1,547, $928) | $247/year (2, $247, $124) | $897/year (4, $495, $536) |
| Telephone | $62/year (1, $62, -) | $12/year (1, $12, -) | $37/year (2, $37, $25) |
| Meal | $62/year (1, $62, -) | $309/year (2, $309, $62) | $227/year (3, $247, $90) |

was extreme, in terms of stress and fear (98% selected extreme level), expenses related to JE (97%), the income of household members (95%), and missing school (89%).

## Discussion

This study is the first to explore JE costs in acute and sequelae stage for up to 10 years following hospital discharge in Laos and Vietnam. This study investigated the cost of illness, including direct medical cost, direct non-medical cost, indirect cost, and other family impact in a total of 242 households with JE patients in Vietnam and 65 households with JE patients in Laos.

The majority of households in both countries resided in rural and peri-urban areas where an average of only 1–2 family members earned an income for a household with 5–6 people. Since JE disease is caused by a mosquito-borne flavivirus, people living in rural areas with poor hygienic and living conditions are usually more susceptible to the disease. Twenty-nine percent of the patients in Vietnam had received a full JE vaccination with three doses. However, it is noteworthy that these patients acquired JE an average of 4.9 years after completing the JE vaccination series. Although these vaccinated patients had more likelihood of mild and moderate levels of severity compared to those who were not vaccinated, even these milder sequelae could be prevented by boosting JE vaccine 3–5 years after completing the primary vaccination with inactivated vaccines [24]. In Laos, about 20% of participants were unaware of their vaccination status and nearly 80% had not been vaccinated. This result has strong implication for the role of JE vaccine on prevention of the illness and its tragic consequences for patients and families.

Direct medical costs were the main contributor to the burden of the disease in all groups, accounting for about 62% of the total costs and up to about 84% for acute patients when they were in hospitals. Costs for medicines and for provider fees (hospitals, doctor office) are the main cost drivers in acute and sequelae care. Compared to the acute group, the direct medical and non-medical costs for JE of the initial rehabilitation and long-term sequelae cases are remarkably lower, which is similar to findings from studies in Nepal and Cambodia [25, 26]. However, the fact that about 75% of cases in Vietnam and 89% of cases in Laos did not seek care despite the patients' sequelae status could drive the low average cost of sequelae care. The average annual costs were three to nine times higher when the calculation only included those who accessed sequelae care. The direct cost of the sequelae groups was mainly spent on brain supplements or transportation expenses for spontaneous doctor visits.

Our estimation on the direct medical cost of care for acute JE and sequelae is comparable to treatment costs reported in a cost-effectiveness study of JE vaccination in Shanghai and

Guizhou, China [27, 28]; but higher than the measures in Cambodia which had a possibility of underestimation as long-term cost was not extrapolated beyond a three-month period [26] and in Bali, Indonesia where treatment costs for JE-associated disability were based on experts' opinion [29]. Our findings thus can expand the evidence to inform JE vaccination cost-effectiveness, which was found in previous studies as cost-saving or highly cost-effective with similar or lower cost of illness [26–30].

The research team evaluated indirect costs through the calculation of actual income loss reported by households and intangible costs, namely school absenteeism and missed working days. Similar to the findings on direct costs, the indirect economic burden was most heavy in acute patients. Families with acute patients in Vietnam reported a total income loss of $1,014 and a considerably long school and work absenteeism of 61.6 days for patients and 73.2 days for other household members, respectively. In Vietnam, sequelae groups had no school absenteeism, but some families had missed working days and lost income for seeking care or taking care of the patient, with a median of income loss registering 0. We documented no data on the school absenteeism and missed working days of the initial rehabilitation and long-term sequelae groups in Laos. However, these results should be viewed within the context that we collected data from the 90 days prior to the date of the interview, and by that time, the patients and their families might have managed to adapt to the circumstances of ongoing JE sequelae, resulting in lower reported impact on school and work.

The impact of JE on families, nevertheless, is not just during the points of care but is spread over years. Even by the date of the interview in 2021–2022, some families were still struggling, either to get a loan to cover the costs of their acute JE family member, or to pay the debt years after the acute JE phase passed. It is also important to note that many families borrowed money from lenders with very high interest rates—up to 45% per month—that could be considered usurious and that create long-term burden with debts from more than $3,000 in the first two years to more than $8,000 in the 10 years after discharge. Considering the 2021 GDP per capita of $3,694 in Vietnam and $2,551 in Laos [31], these loan payback amounts are catastrophic to the families. Most families mentioned that JE had a high to substantial impact on the income of household members and schooling of children suffering from JE. Expenses related to the disease were also of great concern to the families and were the cause of considerable stress and fear.

The results of this study should be considered in the context of some limitations. First, the data were mainly self-reported by respondents and recall bias is unavoidable. However, we extracted clinical and expense data from hospitals to put forward evidence from the hospital perspective and supplement the household data. The study was conducted during the COVID-19 pandemic when most participants could not have a follow-up visit within 90 days. Although we asked the participants to compare their current access to care with pre-pandemic access, our estimation on expenses for re-examination of JE patients at hospitals might not fully reflect the counterfactual situation of non-pandemic. In addition, the micro-costing approach with tedious questionnaires could have prevented in-depth exploration of some direct non-medical expenses, such as costs for ritual offerings, gifts for health staff, etc. The study did not investigate the type of JE vaccine that the patients had received, but according to Vietnam's National EPI, the JE vaccine used locally is the inactivated mouse brain–derived vaccine (JEVAX). It is possible some patients could have received an alternative vaccine such as IMOJEV. Laos uses live attenuated SA 14-14-2 vaccine, but participants from Laos either weren't vaccinated, or they reported not knowing their vaccination status. We did not collect the time interval from last vaccination to the onset of illness, but our analysis of the recommended vaccination schedule suggests considering booster doses to increase vaccine protection after 3–5 years, especially in JE-endemic areas using inactivated vaccines, which do not prompt full immunologic

memory. Lastly, the cases in Laos were drawn from the tertiary hospital database, which may mean that the proportion of each severity level may not be reflective of JE severity in the community.

## Conclusion

Although JE vaccine has long been introduced in Laos and Vietnam, this study shows that JE patients and families still suffer extreme medical, economic, and social hardship where vaccination coverage is low or protection has waned. This calls for improved policies to bolster JE vaccination coverage, increase vaccination regime completion, and encourage booster doses in JE-endemic areas where immune memory may wane 3–5 years after the use of inactivated vaccines. These simple, yet hugely impactful, policy updates will prevent more families from facing substantial impacts of this vaccine-preventable disease.

## Supporting information

**S1 Table. Estimated sample size by stage of illness.**
(DOCX)

**S1 File. Supporting data for Laos.**
(CSV)

**S2 File. Supporting data for Vietnam.**
(CSV)

## Acknowledgments

We would like to express our gratitude to the study participants and families in Laos and Vietnam for their involvement in this study. We acknowledge Ashley Latimer from PATH for the review and important inputs to the final manuscript; Dr. Pham Van Khang and Dr. Tong Thi Thu Ha from the National Institute of Hygiene and Epidemiology; Ms. Nguyen Thi Huyen from Pasteur Institute Ho Chi Minh City; Dr. Pham Thi Lan Lien from Vietnam National Children Hospital; and Ms. Vayouly Vidhamaly, Dr. Xaykhamphet Phommavanh, Dr. Khanxayaphone Phakhounthong, and Mrs. Anisone Chanthongthip from Lao-Oxford University-Mahosot Hospital-Wellcome Trust Research Unit for helping in data collection, processing the samples, and administrative coordination. We deeply thank Prof. Tran Nhu Duong in the National Institute of Hygiene and Epidemiology, Prof. Do Thi Thanh Toan in Hanoi Medical University, and colleagues in Provincial Centers of Disease Control and Prevention in Vietnam and Laos for your support in the study oversight.

## Author Contributions

**Conceptualization:** Rose Slavkovsky, Anthony A. Marfin, G. William Letson, Huong Minh Vu, Clint Pecenka.

**Data curation:** An Le Thanh Nguyen, Hai Thanh Phan, Huong Thi Thu Nguyen, Souphaphone Vannachone, Dang Hai Le.

**Formal analysis:** An Le Thanh Nguyen, Hai Thanh Phan, Huong Thi Thu Nguyen, Souphaphone Vannachone, Dang Hai Le, Son Thai Dinh, Dung Chi Tham, Mayfong Mayxay, Elizabeth A. Ashley, Thai Quang Pham, Clint Pecenka.

**Funding acquisition:** Clint Pecenka.

**Investigation:** Hai Thanh Phan, Huong Thi Thu Nguyen, Souphaphone Vannachone, Dang Hai Le, Audrey Dubot-Pérès, Manivanh Vongsouvath.

**Methodology:** An Le Thanh Nguyen, Rose Slavkovsky, Son Thai Dinh, Anthony A. Marfin, G. William Letson, Huong Minh Vu, Dung Chi Tham, Mayfong Mayxay, Elizabeth A. Ashley, Thai Quang Pham, Clint Pecenka.

**Project administration:** An Le Thanh Nguyen, Rose Slavkovsky, Huong Thi Thu Nguyen, Souphaphone Vannachone, Audrey Dubot-Pérès, Manivanh Vongsouvath, Dung Chi Tham.

**Software:** Hai Thanh Phan.

**Supervision:** An Le Thanh Nguyen, Son Thai Dinh, Dung Chi Tham, Mayfong Mayxay, Elizabeth A. Ashley, Thai Quang Pham, Clint Pecenka.

**Validation:** An Le Thanh Nguyen, Huong Thi Thu Nguyen, Souphaphone Vannachone.

**Writing – original draft:** An Le Thanh Nguyen, Clint Pecenka.

**Writing – review & editing:** An Le Thanh Nguyen, Rose Slavkovsky, Hai Thanh Phan, Huong Thi Thu Nguyen, Souphaphone Vannachone, Dang Hai Le, Audrey Dubot-Pérès, Manivanh Vongsouvath, Son Thai Dinh, Anthony A. Marfin, G. William Letson, Huong Minh Vu, Dung Chi Tham, Mayfong Mayxay, Elizabeth A. Ashley, Thai Quang Pham, Clint Pecenka.

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
