## [Decision Letter · Decision Letter 0]

6 Mar 2023

PGPH-D-23-00078

Estimating the cost of illness of acute Japanese encephalitis and sequelae care in Vietnam and Laos: a cross-sectional study

Dear Dr. Nguyen,

Thank you for submitting your manuscript to PLOS Global Public Health. After careful consideration, we feel that it has merit but does not fully meet PLOS Global Public Health’s publication criteria as it currently stands. Therefore, we invite you to submit a revised version of the manuscript that addresses the points raised during the review process.

We look forward to receiving your revised manuscript.

Kind regards,

Habib Hasan Farooqui, MBBS, MD

Academic Editor

Journal Requirements:

1. You indicated that you had ethical approval for your study. In your Methods section, please ensure you have also stated whether you obtained consent from parents or guardians of the minors included in the study or whether the research ethics committee or IRB specifically waived the need for their consent.

2. In the online submission form, you indicated that "The data that support the findings of this study are available on request from the corresponding author". All PLOS journals now require all data underlying the findings described in their manuscript to be freely available to other researchers, either 1. In a public repository, 2. Within the manuscript itself, or 3. Uploaded as supplementary information.

Additional Editor Comments (if provided):

Dear Authors,

Please refer to the the reviewers comments and please provide a revised version with point wise response to the reviewers comments.

Reviewer 1

The efforts of the authors are highly appreciated in terms of data collection, analysis and calculation of cost of treatment of disease as well as social cost including various non-medical indirect costs. Following are some of the observation require more clarification.

* What kind of gifts are included in the cost estimation which form part of miscellaneous cost?

* The amount of total costs in Table 3 & Table 4 are mot matching total of other components in the same table. Please explain why these total are different.

* The linkage of the cost calculation and vaccination status is not clearly established. The study focuses mostly on OOP and societal cost of care and conclusion directly linking vaccination with cost is misleading.

The work done by the team is of high standards but the objective of the study and conclusion needs further data and analysis to arrive at the findings. Limiting this study to impact of JE on health system would be advisable rather than linking it directly to vaccine coverage.

Reviewer 2

I have received your paper titled: "Estimating the cost of illness of acute Japanese encephalitis and sequelae care in Vietnam and Laos: a cross-sectional study."

General comments

This is a good paper on a topical issue that may inform policymakers in improving Japanese encephalitis vaccination coverage but may benefit from further review considering the comments below.

Specific comments

ABSTRACT

Under the background rearrange the section by mentioning the burden of the disease across the region and state how the aim of the study. Move the perspective of the study to the methodology section.

Under the methodology section, the author can state the perspective, the unit cost, and the type of cost estimate of the study.

Under the results section, it would also be great to present the annual case for initial sequelae care and long-term sequelae care for Laos separately.

INTRODUCTION

To show the burden of care by households and the overall health financing system. The research could provide the percentage of out-of-pocket and health insurance expenditures in Laos from previous studies.

METHODS

It would be great to include a 'unit' cost typology to summarize the resource use measurement of the study, for example, including intervention 'unit' costs, direct and ancillary service 'unit' costs, activity costs, and input costs in the topology.

The analysis section discusses the unit costs for medical costs charged by providers, which were assumed to cover overhead costs. The resource use measurement and the methods used to capture the quantities of resources need to be clarified having applied the micro-costing approach. The research could consider providing a resource measurement section.

RESULTS

The results section only shows detailed results of Vietnam's direct non-medical costs for patients who access care in Vietnam and fails to present the breakdown costs for Laos. The researcher could provide a summarized cost for the different health perspectives, i.e., the health system and the household, as per the study perspective. This will enable easy comparison and also present the results per the study's main aim.

DISCUSSION

How could the study be compared to other regional studies, especially on the cost of implementing vaccination programmes? The researcher could consider referring to a cost-effectiveness study done on the JEVAX to affirm the conclusion of expanding the vaccine coverage.

Reviewers' comments:

Reviewer's Responses to Questions

**Comments to the Author**

1. Does this manuscript meet PLOS Global Public Health’s publication criteria? Is the manuscript technically sound, and do the data support the conclusions? The manuscript must describe methodologically and ethically rigorous research with conclusions that are appropriately drawn based on the data presented.

Reviewer #1: No

Reviewer #2: Yes

2. Has the statistical analysis been performed appropriately and rigorously?

Reviewer #1: Yes

Reviewer #2: Yes

3. Have the authors made all data underlying the findings in their manuscript fully available (please refer to the Data Availability Statement at the start of the manuscript PDF file)?

Reviewer #1: Yes

Reviewer #2: Yes

4. Is the manuscript presented in an intelligible fashion and written in standard English?

Reviewer #1: Yes

Reviewer #2: Yes

5. Review Comments to the Author

Reviewer #1: The efforts of the authors are highly appreciated in terms of data collection, analysis and calculation of cost of treatment of disease as well as social cost including various non-medical indirect costs. Following are some of the observation require more clarification.

* What kind of gifts are included in the cost estimation which form part of miscellaneous cost?

* The amount of total costs in Table 3 & Table 4 are mot matching total of other components in the same table. Please explain why these total are different.

* The linkage of the cost calculation and vaccination status is not clearly established. The study focuses mostly on OOP and societal cost of care and conclusion directly linking vaccination with cost is misleading.

The work done by the team is of high standards but the objective of the study and conclusion needs further data and analysis to arrive at the findings. Limiting this study to impact of JE on health system would be advisable rather than linking it directly to vaccine coverage.

Reviewer #2: The manuscript follows the PLOS Global Public Health's publication criteria, the statistical analysis has been explained and the results presented in table format as per the publication presentation criteria. The author has indicated the availability of data upon request. The manuscript abides by the standard English language and uses cost estimate terms in the study.

6. PLOS authors have the option to publish the peer review history of their article (what does this mean?). If published, this will include your full peer review and any attached files.

**Do you want your identity to be public for this peer review?** For information about this choice, including consent withdrawal, please see our Privacy Policy.

Reviewer #1: No

Reviewer #2: No

---

## [Editor Report · Decision Letter 1]

22 May 2023

Estimating the cost of illness of acute Japanese encephalitis and sequelae care in Vietnam and Laos: a cross-sectional study.

PGPH-D-23-00078R1

Dear Ms Nguyen,

We are pleased to inform you that your manuscript 'Estimating the cost of illness of acute Japanese encephalitis and sequelae care in Vietnam and Laos: a cross-sectional study.' has been provisionally accepted for publication in PLOS Global Public Health.

Best regards,

Habib Hasan Farooqui, MBBS, MD

Academic Editor